# Half-Ring Microlasers Based on InGaAs Quantum Well-Dots with High Material Gain

Fedor Zubov [1,*], Eduard Moiseev [2], Mikhail Maximov [1], Alexander Vorobyev [1], Alexey Mozharov [1], Yuri Shernyakov [3], Nikolay Kalyuzhnyy [3], Sergey Mintairov [3], Marina Kulagina [3], Vladimir Dubrovskii [4], Natalia Kryzhanovskaya [2] and Alexey Zhukov [2]

[1] Center of Nanotechnology, Alferov University, St. Petersburg 194021, Russia
[2] International Laboratory of Quantum Optoelectronics, HSE University, St. Petersburg 190008, Russia
[3] Ioffe Institute, St. Petersburg 194021, Russia
[4] Faculty of Physics, St. Petersburg State University, St. Petersburg 199034, Russia
[*] Correspondence: fedyazu@mail.ru

**Abstract:** We report on half-ring lasers that are 100–200 μm in diameter and are fabricated by cleaving the initial full rings into halves. Characteristics of the half-ring and half-disk lasers fabricated from the same wafer are compared. The active area of the microlasers is based on the quantum heterostructures of mixed (0D/2D) dimensionality, referred to as quantum well-dots with very high material gain. Half-ring lasers show directional light emission and single-mode lasing near the threshold. A maximal continuous-wave output power of 76 mW is achieved for a half-ring that is 200 μm in diameter. Half-rings demonstrate better wall-plug efficiency as compared to half-disks. Lasing in pulse mode is observed up to 140 °C, the characteristic temperature is 100–125 K, depending on the half-ring size. P-side down bonding onto Si-board significantly improves power and temperature characteristics. In CW mode, lasing is maintained up to 97 °C, limited by active-area overheating.

**Keywords:** diode microlasers; half-rings; half-disks; microdisks; quantum well-dots





## 1. Introduction

Presently, miniaturization is one of the main trends in optoelectronics. Scaling down the device size not only enables compact modules, but also reduces power consumption and opens new opportunities for various applications. For instance, compact and energy efficient microlasers are promising as light sources for future short-distance optical communications [1], sensors [2], etc. Microdisk and microring lasers have attracted great attention in the past few decades due to their simplicity of epitaxial growth and subsequent post-growth fabrication [3]. However, the emission from such microlasers is azimuthally isotropic and significantly limits their applications. Directional light output can be achieved by using so-called D-shaped cavities [4] and half-disks [5–7]. Similar to an ideal disk, in a half-disk laser the light wave is travelling along the rim of the cylindrical wall of the resonator and is experiencing total internal reflection. In [5], it was shown that half-disk resonators support optical whispering gallery modes.

Emission in a wide angle and single-mode lasing was demonstrated in a AlGaInAs/InP microdisk laser 16 μm in diameter, with a cleaved flat side [8]. However, lasing was observed only below 10 °C and the output power was relatively low—13 μW. The mode structure of D-shaped and half-disk cavities were thoroughly studied both experimentally and theoretically [9–11]. It was shown that the displacement of a flat facet from the axis can significantly modify optical mode structure. However, to the best of our knowledge, half-ring lasers were studied only in [12]. A threshold current as low as 14.5 mA and an output power of about 1 mW were achieved for a GaAs/AlGaAs quantum well half-ring with a radius of 100 μm. A frequency response extending to 6.5 GHz was demonstrated. In this paper, we present experimental studies of the main characteristics (threshold, power,

spectral and temperature characteristics, as well as radiation directivity) of half-ring lasers of various diameters with active region, based on InGaAs/GaAs quantum well-dots, and we compare their characteristics with those of the half-disks.

## 2. Device Fabrication and Experimental Techniques

The laser heterostructure was grown by metal-organic vapor-phase epitaxy (MOVPE) on an $n^+$-GaAs substrate misoriented by $6°$ towards [111] direction. The laser active region was based on five layers of quantum-well dots (QWDs). Each QWD layer was formed by the deposition of 8 monolayers of $In_{0.4}Ga_{0.6}As$. The GaAs spacers between the QWD layers were 40 nm thick. The laser's double heterostructure comprised a 0.78 μm-thick GaAs waveguide and 1.5 μm thick p- and n-type doped $Al_{0.4}Ga_{0.6}As$ claddings. Peculiarities of epitaxial synthesis, and the optical and laser properties of QWDs are described in detail elsewhere [13]. The advantage of QWDs as an active region for microlasers is that they provide a very high material gain of $1.1 \times 10^4$ cm$^{-1}$ [14], which makes it possible to achieve lasing in very small cavities with high output optical losses.

Three types of microlasers with outer diameters of 100, 150 and 200 μm were fabricated by optical lithography, dry etching and cleaving: half-disks with half-round contact (HDs, Figure 1a), half-disks with half-ring contact (HDRs, Figure 1b) and half-rings with half-ring contact (HRs, Figure 1c,d). The inner radii of HR mesas were 19 μm smaller than the outer ones. The mesas were etched down through the bottom cladding to the depth of about 6.5 μm. Half-ring contacts of HDRs and HRs had the width of 15 μm and were separated from the mesas' edges by 2 μm. Offsets of half-round contacts from HDs' edges were also 2 μm. Half-disks and half-rings were formed by the precise cleaving of the chips with the full disks and full rings along their diameters. GaAs-chips with the microlasers were soldered n-side down onto Cu-heatsinks. Some devices were bonded p-side down onto Si-boards, on which metal tracks were preformed, providing individual pumping of microlasers. Details of half-disk fabrication are presented in our previous papers [6,7].

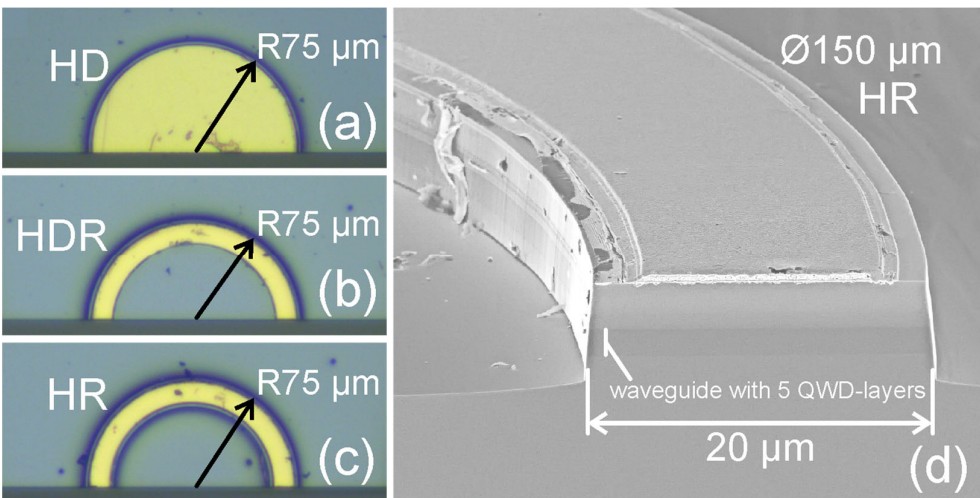

**Figure 1.** Optical microscopy images of a half-disk with half-round contact (HD, (**a**)), half-disk with half-ring contact (HDR, (**b**)), half-ring with half-ring contact (HR, (**c**)) and scanning electron microscopy image (**d**) of an HR cleaved edge. The diameters of all devices are 150 microns.

The measurements were carried out under continuous wave (CW) and pulse pumping at various temperatures of heatsinks. Electroluminescence (EL) spectra were acquired with a Yokogawa AQ6370C optical spectrum analyzer with 20 pm spectral resolution. The absolute value of the optical power was measured with a Thorlabs FDG1010 1 cm × 1 cm photodiode placed close to the microlasers under test ($\approx 136°$ collection angle). Far fields were recoded with a CCD camera and a goniometer equipped with a photodiode ($0.3°$ angular resolution). Due to the wide collection angle of the photodiode, the measured optical power was more than 90% of the total output power of the radiation emitted from the cleaved facets (this

value was estimated based on far field measurements using the goniometer). Although devices had some imperfections (debris on mesas' sidewalls formed during the etching step, and some variations in the metal contact dimensions, see Figure 1d), the lasers showed insignificant scattering in parameters for each design type and diameter. Below, we present the results for the best devices.

## 3. Results and Discussion

### 3.1. Light–Current Characteristics

The light–current (L-I) curves of Ø200 μm microlasers under study are compared in Figure 2a. The differential efficiencies of the HR and HDR are higher than that of the HD. In microdisk lasers, circular symmetry and total internal reflection of light result in poor light outcoupling. The output of radiation from a microdisk is mainly caused by the light scattering on the sidewall roughness. However, such roughness is typically insignificant in providing efficient light extraction, and if a scatterer (notch, microsphere, etc.) is not specially formed on a disk surface [15], the output power is small. In half-disks and half-rings the optical losses are mainly associated with the light emission through cleaved facets (similar to the output losses of a stripe laser). These losses are high enough owing to short cavity length (from facet to facet) and, correspondingly, the output power can be much higher than in microdisks. The internal optical losses and internal differential quantum efficiency determined using stripe lasers fabricated from the same heterostructure were 1.5 cm$^{-1}$ and 0.82, respectively [13].

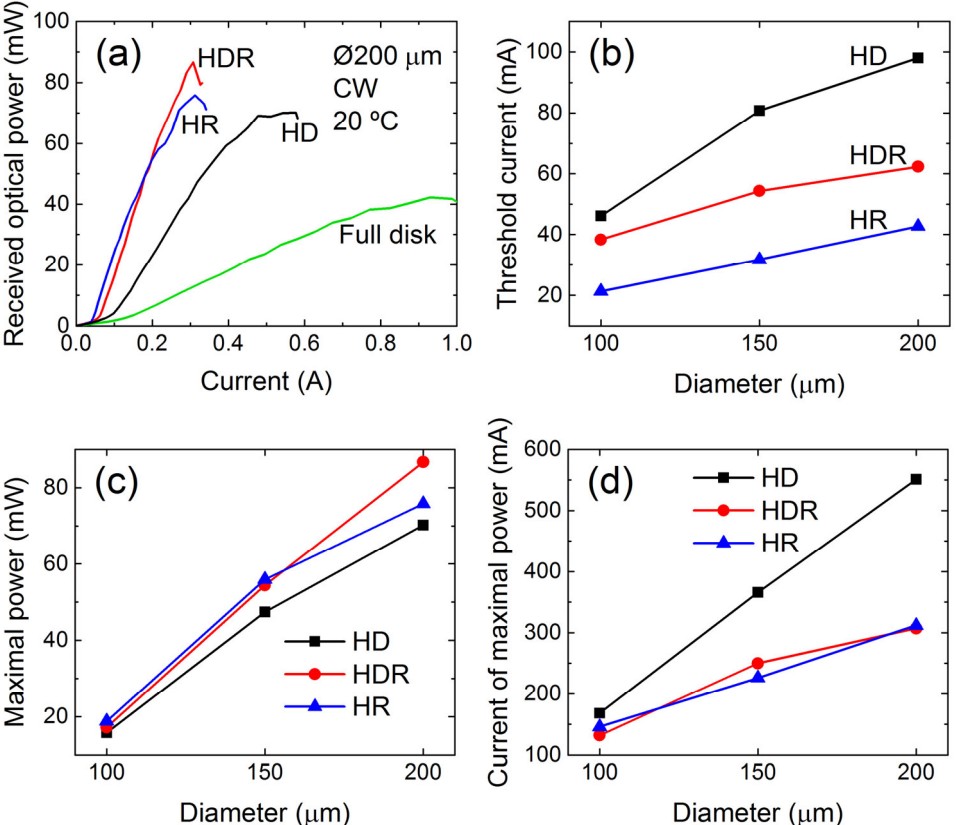

**Figure 2.** The light–current curves (**a**), threshold currents (**b**), maximal optical powers (**c**) and currents at which maximal powers are achieved (**d**) of HDs, HDRs and HRs of various diameters. Panel (**a**) also presents data for a Ø200 μm full disk with round contact for comparison. Data correspond to 20 °C CW measurements of p-side-up mounted devices.

The output optical losses $\alpha_m$ of half-disk/ring cavities due to emission from cleaved facets can be estimated using the well-known expression for stripe lasers: $\alpha_m = -\ln(R)/L$, where $R$ is the reflection coefficient of the cleaved facet and $L$ is the cavity length. In the

case of half-disk/ring resonators, *L* is approximately equal to the perimeter of the rounded part of the resonator. The reflection coefficient *R* of an AlGaAs cleaved facet is about 0.3. Then, for half-disks/rings with diameters of 100, 150 and 200 μm, the output optical losses at cleaved facets can be estimated as 64, 42 and 32 cm$^{-1}$, respectively. It was shown in [13] that the modal gain of one layer of QWDs can reach 75 cm$^{-1}$.

We attribute higher efficiency of the HR and HDR to the fact that the current is injected only (or at least predominantly) in the device periphery, where high-Q lasing modes with low radial numbers propagate and, therefore, current is not wasted in pumping the central non-lasing part. In real half-disks, the optical mode structure can be more complex, but the most intensive optical modes still propagate closer to the device periphery [10]. In HRs, only modes with low radial numbers exist because the central part is etched out. Contact geometry of HDRs provides predominant pumping of optical modes with low radial numbers propagating under half-ring contact. The carrier diffusion and self-absorption can lead to moderate pumping of some optical modes, with high radial numbers located in the central part of HDRs. However, in the case of the QWD active region, the carrier-lateral transport is suppressed and the pumping of optical modes in the central part of HDRs is inefficient.

It was found that the HR-type devices have the lowest threshold currents for all diameters we studied (Figure 2b). Additionally, although the maximal optical power does not vary significantly for devices of different types with the same diameter (Figure 2c), the current at which it is achieved is significantly lower in the HDRs and HRs compared to the HDs (Figure 2d). For instance, for the HD that is 200 μm in diameter, it is about 75% larger than in the HDR and HR.

For full disks having a round contact, HDs and HRs one can assume a uniform current distribution over mesas. Then the threshold current densities of the full disk, HD and HR of 200 μm in diameter can be estimated as 404, 615 and 790 A/cm$^2$, respectively. The threshold current density for the HDR cannot be estimated with reasonable precision, since the exact current spreading outside the ring contact and the corresponding current density are not known for this type of device. The full disk has the lowest threshold current density as it is characterized by the smallest output optical losses. The higher threshold current density for the HR as compared to the HD can be attributed to the additional nonradiative recombination at the inner sidewall of the mesa. We note that the current density at which the thermal roll-over occurs in the HR (about 5.8 kA/cm$^2$) is almost two times higher than that in the full disk and HD (about 3.0 kA/cm$^2$). This indicates improved heat removal in the HR type device, which is consistent with the results of temperature studies presented below.

The fact that only high-Q optical modes are pumped, and the current is not consumed for the pumping of non-lasing modes, results in about a 70% increase in the wall-plug efficiency (WPE) of the Ø200 μm HR and HDR as compared to the HD (Figure 3a,b). The highest WPE is achieved in the Ø200 μm HR and amounts to 15.7%. The power-conversion efficiency of the HRs is slightly higher than that of the HDRs and reaches its maximal values at lower injection currents, which is observed for all diameters (Figure 3b,c). Predictably, full disks exhibit the lowest efficiency among the studied microlasers (Figure 3a). Slightly lower efficiencies in the HDRs, achieved at higher currents as compared to the HRs, are attributed to a higher threshold current and current spreading outside the area under the ring contact towards the central part of the devices. The maximal efficiency for all types of microlasers drops with a decrease in their diameter (Figure 3b), which we attribute to an increase in the internal optical losses. Another interesting observation is that for 200 μm diameter, the optical power at the maximal WPE (Figure 3d) in the HDR is much higher than in the HR (62 and 36 mW, respectively), while the maximal WPE of these devices is approximately the same (15.2 and 15.7%, respectively).

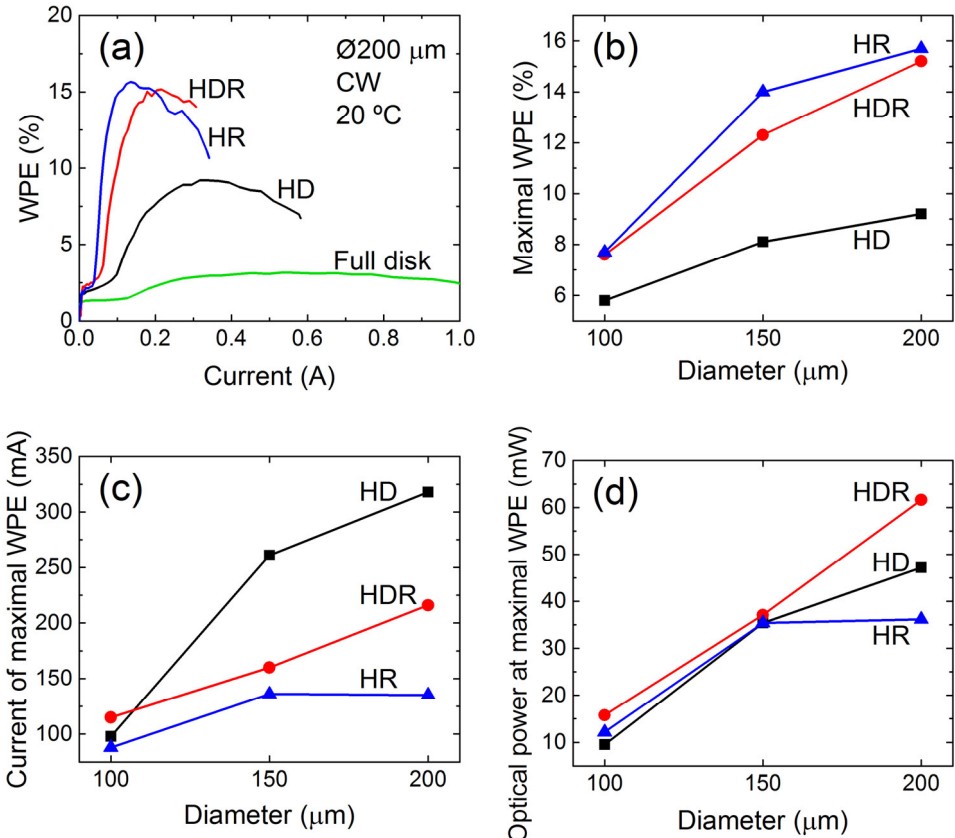

**Figure 3.** The wall-plug efficiency (WPE) versus the injection current (**a**), maximal WPE (**b**), currents at which maximal WPE is achieved (**c**) and optical powers corresponding to maximal WPE (**d**) of HDs, HDRs and HRs of various diameters. Panel (**a**) also presents data for a Ø200 μm full disk with round contact for comparison. Data correspond to 20 °C CW measurements of p-side-up mounted devices; the WPE corresponds to the optical power received by photodiode.

### 3.2. Effect of Microlaser Bonding

The maximal output power of all microlasers is limited by the thermal roll-over due to the active region overheating (Figure 2a). P-side-down bonding of microlasers onto Si-boards improves the heat removal, since the distance between an active region and a heatsink is much smaller compared to the p-side-up mounting discussed above. This should enable the increase in the output power and improve the temperature performance (discussed below). The effect of the bonding should be especially pronounced for a Ø100 μm microlaser because it has the smallest footprint [16]. The LI-curves for p-side-up soldered and p-side-down bonded HRs are compared in Figure 4a. The bonding results in an increase in the differential efficiency and a 1.9-fold enhancement in the maximal output power (up to 35.5 mW). For the bonded HR-lasers, the thermal roll-over takes place at a 56% higher injection current (at 228 mA) and the maximal WPE after the bonding increases from 7.7 to 11.2%.

### 3.3. Spectral Characteristics and Far Fields

Emission spectra in nominally identical HR lasers vary slightly from one device to another. HRs show quasi-single mode lasing with side-mode suppression ratio (SMSR) of more than 20 dB in a certain range of injection currents. An example of the quasi-single mode lasing of a HR is presented in Figure 4b. However, some HRs exhibited multimode lasing right above the threshold. We attribute the occurrence of the quasi-single mode lasing to a slight (by 1–3 μm) deviation in the position of a cleaved facet with respect to the center, which, as it was shown for the case of half-disks in [10], can lead to the suppression of side modes. However, more detailed studies of the impact of microlaser geometry on

mode structure are needed to gain deeper understanding of this issue. We believe that further cavity optimization (the position of the cleavage plane, the half-ring mesa width and diameter) will make it possible to achieve higher SMSR.

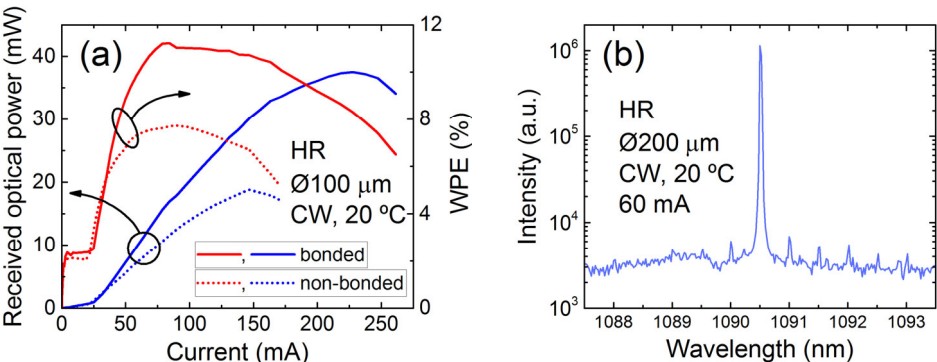

**Figure 4.** (**a**) The light–current curves and wall-plug efficiencies (WPEs) versus the injection current of p-side-up mounted (non-bonded) and bonded onto Si-board Ø100 μm HRs. (**b**) The lasing spectra of a Ø200 μm HR at 60 mA (the threshold current—42.7 mA).

Lasers with half-disk cavities provide better directionality of the light output as compared to full disks [6]. As in [6], we compared the optical power received by the large area photodiode (≈136° collection angle) emitted from cleaved edges of the HRs (front power) and from their circular parts (back power). An example of such a comparison is shown in Figure 5a for the Ø150 μm HR. A dramatic difference in front and back maximal optical powers is found: their ratio is about 18.6. For the HRs with diameters of 100 and 200 μm, this ratio is 8.5 and 15.5, respectively.

To study far field patterns of the HRs qualitatively, we performed measurements with the CCD camera. A pattern formed on a CCD sensor by the radiation emitted from both edges of the Ø100 μm HR in the lasing mode is shown in Figure 5b. The collection's angles in the center parts of the sensor are about 118 and 103 degrees in lateral and vertical directions, respectively. It is seen that the frontal far field pattern is rather broad and exhibits interference fringes formed by beams emitted from the left and right edges of the HR. The pattern significantly changes with an increase in the injection current, which we attribute to sequential excitation of different optical modes. Similar patterns are observed for devices of all types and diameters. As the diameter increases, the interference fringes become denser.

To measure far fields emitted only by one HR edge we masked another edge by a metal foil. The far field pattern of the Ø200 μm HR in lasing mode obtained by the goniometer is presented in Figure 5c. No interference fringes are seen. Using corresponding profiles in the vertical direction at φ = −17.8° for the brightest vertical stripe (Figure 5d) and in the lateral direction at θ = 90° (Figure 5e) we estimate the divergence of the frontal radiation of one edge of the HR: the full width at half maximum (FWHM) of the vertical profile is about 32° and in the lateral direction (in the growth plane) 90% of the optical power is concentrated in the angle of about 120°.

### 3.4. Near Field Patterns

Figure 6 shows images from an optical microscope of a cleaved facet (more precisely, the right half of it) of a half-disk (the HD or HDR) with a diameter of 200 μm in the absence of pumping under external illumination (Figure 6a) and at pumping close to the threshold without external illumination (Figure 6b—the HD and Figure 6c—the HDR), as well as the corresponding distributions of the radiation intensity along the waveguide layer (the near field profiles, Figure 6d). It can be seen from the distributions that both types of devices have a rather complex mode structure (due to multi-mode emission at this pumping). In the HDR, the intensity peaks of the optical modes are concentrated near the edge of the half-disk in a region of about 24 μm (FWHM) under the half-ring contact. For the HD,

which has a solid contact, the brighter emission corresponding to optical modes extends from the edge to the center of the device for a greater distance (its FWHM is 66 μm). The intensity of spontaneous emission (regions without intensity peaks) for the HDR is more than four times lower than for the HD, which indicates the suppression of pumping of the central area of the device with a half-ring contact.

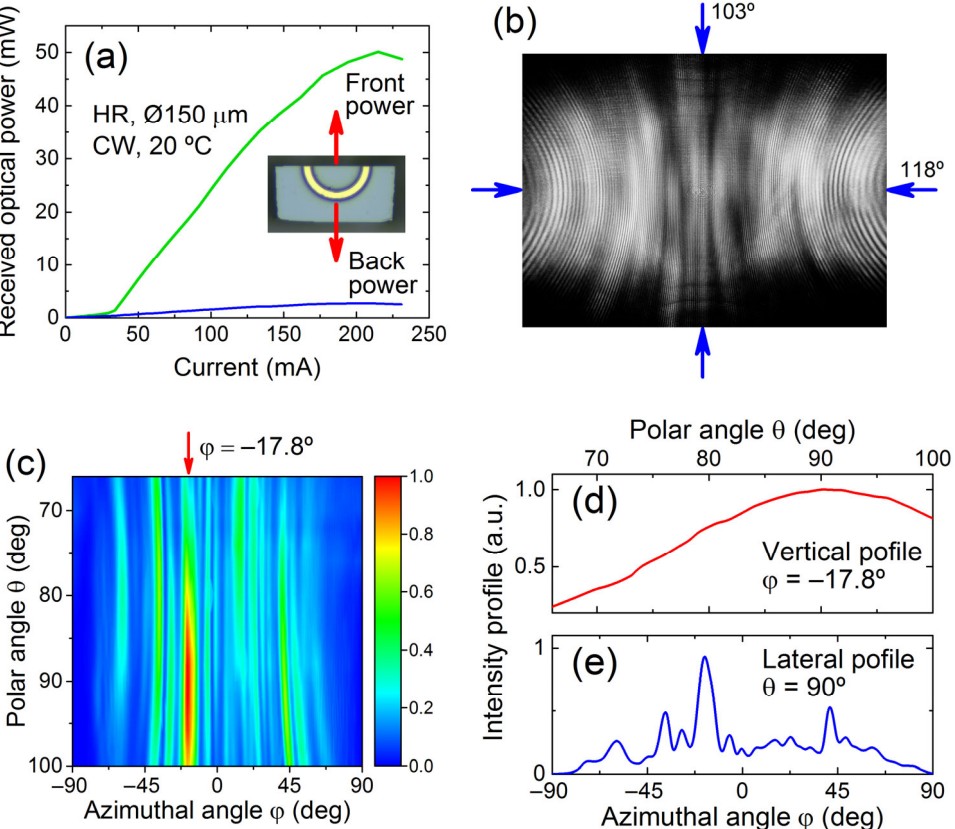

**Figure 5.** (**a**) The light–current curves for radiation emitted from cleaved edges (front power) and circular side (back power) of the Ø150 μm HR. (**b**) A pattern formed on the CCD sensor by the radiation emitted from both edges of the Ø100 μm HR in the lasing mode (the HR center is located opposite the sensor center; the plane of the HR heterolayers is oriented horizontally; approximate vertical and lateral radiation collection angles are marked in the figure). (**c**) Far field pattern of one edge of the Ø200 μm HR in the lasing mode, measured with the goniometer (the normal to cleaved facet of the device corresponds to the azimuth angle $\varphi = 0$ and the polar angle $\theta = 90°$), as well as the corresponding profiles in the vertical direction (**d**) at $\varphi = -17.8°$ (for the brightest vertical stripe) and in the lateral direction (**e**) at $\theta = 90°$ (in the growth plane).

Figure 6 also shows optical microscope images of an edge of the HR with a diameter of 100 μm in the absence of pumping under external illumination (Figure 6e), at the threshold (Figure 6f) and above the threshold (Figure 6g). The corresponding distributions of the radiation intensity along the waveguide layer are presented in Figure 6h. It is seen that near the lasing threshold, the near field profile contains several peaks due to multi-mode emission. However, with an increase in pumping, only one mode begins to prevail in the spectrum, which manifests the onset of quasi-single mode lasing and only one bright spot starts to dominate in the near field (Figure 6h).

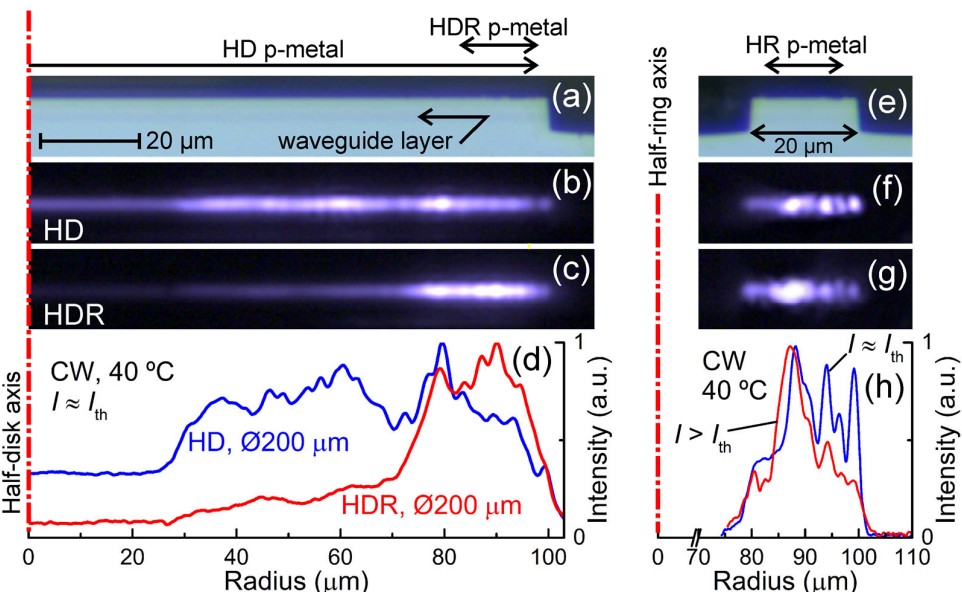

**Figure 6.** Optical microscope images of a cleaved facet of the right half of a Ø200 μm half-disk at zero current under external illumination (**a**), and at the threshold pumping of HD (**b**) and HDR (**c**), as well as the corresponding distributions of the radiation intensity along the waveguide layer (the near field profiles) (**d**). Optical microscope images of an edge of the Ø100 μm HR at zero current under external illumination (**e**), at the threshold pumping (**f**) and above threshold pumping (**g**), as well as the corresponding distributions of the radiation intensity along the waveguide layer (**h**).

### 3.5. Temperature Performance

Temperature dependencies of the threshold current were measured in pulsed and CW modes. The pulse duration of 300 ns and the repetition rate of 4 kHz were chosen to avoid the active region from overheating. The microlasers' temperature characteristics in pulsed mode are determined by active region physical properties (localization energies, density of states, carrier capture and thermal escape mechanisms, radiative and non-radiative recombination, etc.) and not affected by quality/type of mounting. All types of the microlasers show lasing up to 140 °C. The measurements were limited by the melting point of the solder used (InSn). In the temperature range from 20 to 80 °C, the temperature dependence of the threshold current density $j_{th}$ is fairly well described by the well-known equation $j_{th}(T) = j_{th}(20\ °C)\cdot\exp[(T-20\ °C)/T_0]$, where $T_0$ is the characteristic temperature of the threshold current density. The Ø200 μm HR and HD show lower threshold current density and better temperature stability as compared to the Ø100 μm HR and HD (Figure 7). We attribute this to an increase in the output optical losses in half-disks and half-rings with a decrease in the diameter [7], which causes a higher threshold occupation of an active region and waveguide by charge carriers. The threshold current component, corresponding to waveguide recombination, is highly temperature sensitive [17] and deteriorates $T_0$ in devices with higher output losses.

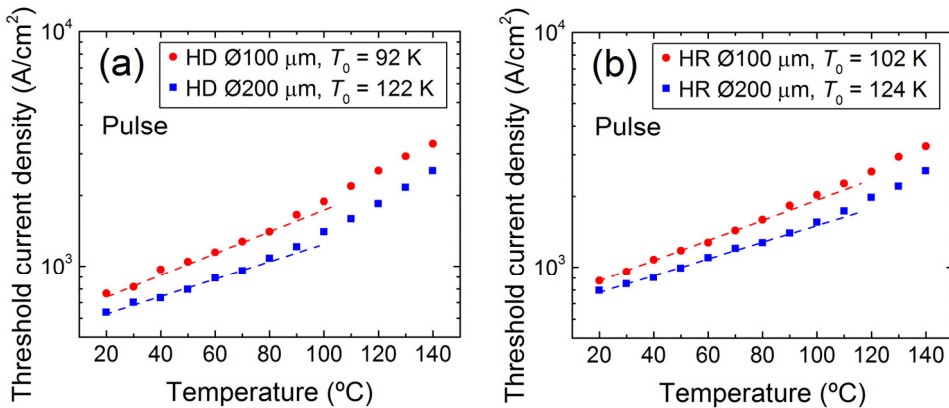

**Figure 7.** Temperature dependence of the threshold current density of the HD (**a**) and HR (**b**) lasers of 100 and 200 μm in diameter in pulsed pumping mode.

The slightly higher $T_0$ in the Ø100 μm HR as compared to the HD can be explained by the higher threshold current density of the HR caused by the contribution of additional sidewall non-radiative recombination at the HR inner hole. This component of the threshold current is nearly temperature-independent and results in some increase in $T_0$. Beyond that, the quality of mesa sidewalls and correspondingly non-radiative recombination rates may vary slightly for different devices due to "technological noise" and result in some scattering in $T_0$.

For the majority of practical applications, the most significant are the temperature properties of the device ($T_0$, the maximal lasing temperature $T_{max}$, lasing line temperature stability, etc.) in CW mode. We also present here the results of a study of the temperature behavior of the devices in a CW pumping regime. An example of temperature dependence of the L-I-curve of the Ø100 μm p-side-down bonded HR is depicted in Figure 8a. Device temperature performance in CW mode is determined by intrinsic temperature behavior (at pulsed pumping discussed above) together with the self-heating. The latter depends on the thermal resistance of a device, which is affected by semiconductor materials used, the geometry of a microlaser and the type/quality of mounting.

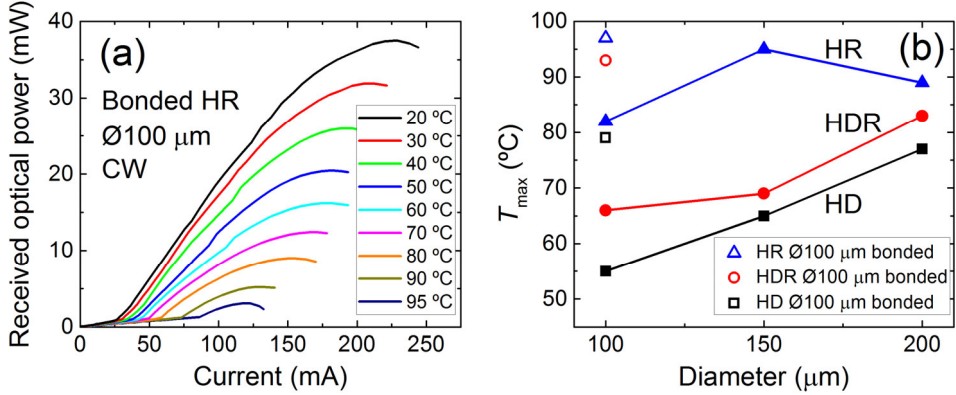

**Figure 8.** (**a**) The light-current curves of a Ø100 μm HR bonded onto a Si-board at various temperatures measured in a CW pumping regime. (**b**) The maximal operation temperature $T_{max}$ of the non-bonded (p-side up mounted) and bonded microlasers versus device diameter.

The maximal operation temperatures $T_{max}$ of all devices under study are compared in Figure 8b. The HRs demonstrate the highest $T_{max}$ and the HDs show the lowest $T_{max}$ for non-bonded and bonded devices of all diameters. Despite the non-monotonic behavior of the $T_{max}$ dependence on device size of the HRs (likely due to a variation in mounting quality) the trend is that the maximal operation temperature increases with microlaser diameter. The bonding significantly improves the heat removal and thus results in an

increase in $T_{\max}$ for bonded microlasers. The maximal lasing temperature is reached in the Ø100 µm HR bonded onto Si-board, and amounts to 97 °C.

The data on the characteristic temperature of the threshold current density in CW mode ($T_0^{CW}$) of all devices are collected in Table 1. As in the case of the maximal operation temperature, the highest values of $T_0^{CW}$ are achieved in devices of the HR type. The highest $T_0^{CW}$ of 104 K detected in p-side up mounted (non-bonded) Ø200 µm HR.

**Table 1.** The characteristic temperature of the threshold current density for CW mode pumping $T_0^{CW}$ (K) of HDs, HDRs and HRs of various diameters and mounting types.

| Diameter, Mount Type | HD | HDR | HR |
|---|---|---|---|
| 200 µm, non-bonded | 80 | 96 | 104 |
| 150 µm, non-bonded | 61 | 70 | 79 |
| 100 µm, non-bonded | 51 | 54 | 79 |
| 100 µm, bonded | 88 | 78 | 78 |

## 4. Conclusions

In conclusion, we fabricated and studied half-disk and half-ring lasers that were 100, 150 and 200 µm in diameter. In such geometry, the laser power is predominantly emitted from the flat side, which is an advantage as compared to full-ring/disk lasers. Another advantage is that such lasers are able to demonstrate single-mode emission without using distributed feedback gratings. The maximal output power of half-ring lasers is within the range of 19–76 mW, which is sufficient for various applications. The highest optical power of 86.7 mW was reached in a Ø200 µm half-disk laser with a half-ring contact. The devices operate up to 140 °C in pulsed mode and up to 97 °C in CW mode. We believe that half-ring/disk lasers are promising for optical data transmission over short distances. The developed devices can be relatively easily integrated onto Si/SOI-boards using p-side-down bonding to create photonic ICs, in which the radiation from a half-disk or half-ring is efficiently coupled to planar waveguide. The radiation from each arm of HRs can be coupled to its own waveguide. Since the emitted beams are coherent, they can be used, e.g., in micro-interferometers, light sources for quantum optics, etc. [18].

**Author Contributions:** Conceptualization, F.Z. and M.M.; methodology, A.V., A.M., M.K., N.K. (Nikolay Kalyuzhnyy) and S.M.; investigation, F.Z., E.M. and Y.S.; data curation, F.Z., E.M. and Y.S.; writing—original draft preparation, F.Z. and M.M.; writing—review and editing, F.Z., M.M. and A.Z.; project administration, M.M. and A.Z.; funding acquisition, M.M., V.D. and N.K. (Natalia Kryzhanovskaya). All authors have read and agreed to the published version of the manuscript.

**Funding:** This work was supported by the Ministry of Science and Higher Education of the Russian Federation under project FSRM-2023-0010 (F.Z. and M.M.). Support of spectral measurements from the Basic Research Program of the National Research University Higher School of Economics is gratefully acknowledged (E.M., N.K. (Natalia Kryzhanovskaya) and A.Z.). V.D. gratefully acknowledges the financial support of the research grant of St. Petersburg State University No. 75746688.

**Institutional Review Board Statement:** Not applicable.

**Informed Consent Statement:** Not applicable.

**Data Availability Statement:** The data presented in this study are available on request from the corresponding author. The data are not publicly available due to the author's readiness to provide it on request.

**Acknowledgments:** The spectral measurements were carried out on the equipment of the unique scientific setup «Complex optoelectronic stand» of HSE University, St. Petersburg.

**Conflicts of Interest:** The authors declare no conflict of interests.

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
