# Peer review of "Half-Ring Microlasers Based on InGaAs Quantum Well-Dots with High Material Gain"

_photonics, doi:10.3390/photonics10030290_

Round 1
Reviewer 1 Report
Please refer to the comments included in the attached review file.

Reviewer 2 Report
1) As the authors pointed out, there has been only one paper [10] in 1990 by Kam Lau’s group regarding the half ring cavity laser. The reason is supposed that no one has not found the way to use this half ring cavity in any application systems due to its drawbacks.
a) There are two outputs in one side and one of them may disturb the system as a stray light.
b) Due to a bending loss, the efficiency may be deteriorated.
2) Even if the authors aim at micro-cavity, there have been various structures such as VCSELs, photonic crystal lasers, membrane lasers and so on that have already engineered.
They should claim the advantage of half-ring cavity over them.
Concludingly, the present reviewer does not agree to publish this paper, if the authors do not show some of clear ways to apply this kind of half ring laser in real systems.
Round 2
Reviewer 1 Report
The authors have nicely addressed many of the original questions and critiques.
Reviewer 2 Report
The authors added sufficient descriptions on this device and the reviewer agrees to publish this paper as the revised version.